# Ultrasound Renal Score to Predict the Renal Disease Prognosis in Patients with Diabetic Kidney Disease: An Investigative Study

**DOI:** 10.3390/diagnostics13030515

**Published:** 2023-01-31

**Authors:** Young Rok Ham, Eu Jin Lee, Hae Ri Kim, Jae Wan Jeon, Ki Ryang Na, Kang Wook Lee, Dae Eun Choi

**Affiliations:** 1Nephrology, Chungnam National University, Daejeon 35015, Republic of Korea; 2Nephrology, Chungnam National University Hospital, Daejeon 35015, Republic of Korea; 3Nephrology, Chungnam National University Sejong Hospital, Sejong 30099, Republic of Korea; 4Medical Science, Chungnam National University, Daejeon 35015, Republic of Korea

**Keywords:** renal ultrasound, diabetes mellitus, renal replacement therapy

## Abstract

Renal disease associated with type 2 diabetes mellitus (T2DM) has become the leading cause of chronic kidney disease (CKD). Renal ultrasonography is an imaging examination required in the work-up of renal disease. This study aimed to identify the differences in renal ultrasonographic findings between patients with and without DM, and to evaluate the relationship between renal ultrasound findings and renal prognosis in patients with DM. A total of 252 patients who underwent renal ultrasonography at Chungnam National University Hospital were included. Kidney disease progression was defined as a ≥10% decline in the annual estimated glomerular filtration rate (eGFR), which, in this paper, is referred to as ΔeGFR/year, or the initiation of renal replacement therapy after follow-up. The renal scoring system was evaluated by summing up the following items: the value of renal parenchymal echogenicity (0: normal; 1: mildly increased; and 2: increased) and the shape of the cortical margin (0: normal and 1: irregular; right kidney length/height (RH—0 or 1), mean cortical thickness/renal length/height (CKH—0 or 1), and cortical thickness/parenchymal thickness (CK/PK—0 or 1) based on the median: 0—above median, and 1—below median). Patients with DM had thicker renal PKH than those without, despite having lower eGFRs (0.91 ± 0.15, 0.86 ± 0.14, *p* = 0.006). In the progression group, the renal scores were significantly higher than those from the non-progression group. In the multivariate logistic regression analysis, the higher renal scores, presence of DM, and younger age were independently predicted for renal disease progression after adjusting for confounding variables, such as the presence of hypertension, serum hemoglobin and albumin levels, and UPCR. In conclusion, patients with high renal scores were significantly associated with renal disease progression. Our results suggest that renal ultrasonography at the time of diagnosis provides useful prognostic information in patients with kidney disease.

## 1. Introduction

The prevalence of type 2 diabetes mellitus (T2DM) is increasing and is one of the greatest healthcare burdens worldwide [1,2,3]. Renal disease associated with T2DM is the leading cause of chronic kidney disease (CKD) in the United States, occurring in 43.5% of patients with T2DM [4], and its prevalence in patients with T2DM is three times higher than that in those without DM [5]. Although a previous study reported that the prognosis of diabetic kidney disease is poorer than that of non-diabetic kidney disease [6], factors associated with poor prognosis have not been fully elucidated.

Renal ultrasonography is a method that is widely used to assess kidney disease because it is noninvasive, safe, and suitable for outpatients, and has low cost, as well as being considerably effective in imaging studies of different structures that constitute the kidney parenchyma [7,8]. It is also used to evaluate acute and chronic renal failure, nephrolithiasis, and hematuria [9]. Among the ultrasound findings, kidney size is important in detecting renal abnormality and predicting renal function [10]. Moreover, a thin, echogenic cortex refers to irreversible damage, whereas a thick, normal echogenic cortex may indicate reversible damage [11].

Several previous studies have detected differences in ultrasound findings between patients with and without DM [12,13,14]. They reported that renal volume and length were larger and longer, while renal cortical echogenicity was higher in patients with early-stage DM than in those without DM. In addition, other studies reported that renal size can be used to predict microalbuminuria and progression in patients with type 1 DM [3,15]. As such, some characteristics of renal ultrasound in patients with DM look different from those without DM, and ultrasound findings in patients with DM are related to the current renal function or proteinuria. However, among these studies, no reports revealed the relationship between renal ultrasound findings and renal prognosis.

We hypothesized that ultrasound findings in patients with DM can be used to predict the prognosis for renal function deterioration. This study aimed to identify the differences in renal ultrasound findings between patients with and without DM, and to create a model that can predict renal prognosis using these findings.

## 2. Materials and Methods

### 2.1. Patient Selection

All patients who had undergone diagnostic renal ultrasonography from January 2013 to December 2015 were included in this study. Exclusion criteria were as follows: (1) estimated glomerular filtration rate (eGFR) of <45 mL/min/1.73 m^2^ at the time of renal ultrasound; (3) the presence of hydronephrosis or single kidney; (4) established contraction of the renal parenchyma; and (5) renal size asymmetry (defined as a difference of ≥2.0 cm). This study was reviewed and approved by the ethics committee of the Chungnam National University Hospital, and was conducted using the Declaration of Helsinki guidelines (IRB No. 2020-04-026).

### 2.2. Clinical Parameters

Baseline data at the time of renal ultrasonography were obtained from medical records and included age, sex, the presence of hypertension (HTN) or DM, and height and weight (measured as meters and kilograms, respectively). Laboratory data for serum creatinine and albumin levels, the estimated glomerular filtration rate (eGFR), and the spot urine protein-to-creatinine ratio (UPCR) were obtained at the outpatient departments. The eGFR was calculated using the CKD Epidemiology Collaboration equation (CKD-EPI).

### 2.3. Study Group Design and End-Points

Patients were divided into two groups depending on their DM diagnoses during ultrasound. In addition, the worsening of renal disease was evaluated based on the GFR. Renal disease progression was defined as a ≥10% decline in the annual eGFR (ΔeGFR/year) or the initiation of renal replacement therapy after follow-up.

### 2.4. Renal Ultrasonography Examinations

All renal ultrasonography was performed by three experienced investigators, with 5 years of experience in the field of renal values in the outpatient department of our renal unit. Examinations were performed in a supine or prone position using standard gray-scale B-mode imaging with 3.5 MHz measurements.

The renal length (RL) was measured as the longest pole-to-pole distance in the sagittal plane measured to the nearest millimeter. In the sagittal plane, the renal cortical thickness (CK, from the medullary pyramid to the capsule) and the renal parenchymal thickness (PK, from the sinus fat to the renal capsule) were measured on the upper, middle, and lower thirds of the kidney, and the average was calculated to avoid any bias due to border variability (Figure 1). To correct the renal size differences based on the patient’s height, the kidney length and parenchymal and cortical thicknesses were measured using the patient’s height [7].

We additionally created a renal scoring system for factors related to prognosis among items observed with renal ultrasound. The renal scoring system was evaluated by summing up the following items: the value of renal parenchymal echogenicity (0: normal; 1:mildly increased; and 2: increased) and the shape of the cortical margin (0: normal and 1: irregular; right kidney length/height (RH—0 or 1), cortical thickness/renal length/height (CKH—0 or 1), cortical thickness/parenchymal thickness (CK/PK—0 or 1) based on the median: 0—above median; 1—below median). The maximum score obtained was 7 points. The calculation process and results from the renal scoring system, for the representative patient, are described in Figure 2 and Table 1.

### 2.5. Statistical Analysis

Comparison among three or more groups was performed using a one-way analysis of variance with the Bonferroni post hoc test. Differences were considered significant at a *p*-value of <0.05.

A comparison of univariate predictors of clinical outcomes between groups was performed using χ^2^ tests for categorical variables and the Kruskal–Wallis or Mann–Whitney tests for continuous variables. Differences in continuous variables between the two groups were assessed using independent *t*-tests and are expressed as means ± standard deviation, and categorical variables are expressed as frequencies and percentages. Multivariate logistic and proportional hazards regression analyses were performed to determine independent variables associated with renal outcomes. All statistical analyses were performed using the statistical software SPSS version 24.0 (SPSS, Chicago, IL, USA). A *p*-value < 0.05 was deemed to indicate statistical significance.

## 3. Results

### 3.1. Clinical Baseline Characteristics and Differences between Patients with and without DM

The baseline demographic and clinical characteristics of 252 patients are shown in Table 2. Among them, 103 (40.9%) had DM and 166 (65.6%) had HTN. The mean follow-up period was 57.4 (range, 24–72) months. The mean serum creatinine level and eGFR were 1.12 (range, 0.52–2.13) mg/dL and 71.1 (range, 40.4–119.5) mL/min/1.73 m^2^, respectively. By the end of the observation period, a total of 33 patients (13.5%) reached the primary renal outcome (29 patients (11.5%) maintaining renal replacement therapy, and 33 patients (13.1%) with >10% decline in annual eGFR). A total of 36 patients had autoimmune diseases, including glomerulonephritis, and the diseases were as follows: IgA nephropathy (*n* = 14), membranous glomerulonephritis (*n* = 6), rheumatoid arthritis (*n* = 6), minimal change disease (*n* = 2), focal segmental glomerulosclerosis (*n* = 2), monoclonal gammopathy of undetermined significance *(n* = 2), IgM nephropathy (*n* = 1), and lupus nephritis (*n* = 1).

Patients with DM were older (*p* = 0.001) and more likely to have HTN and advanced CKD stage at baseline (*p* = 0.000, *p* = 0.000, respectively) than those without DM. Serum hemoglobin levels and eGFR were lower (*p* = 0.000, *p* = 0.000, respectively), in addition to the serum creatinine level and urine protein–creatinine ratio (UPCR), which were higher in patients with DM than those without it (*p* = 0.000, *p* = 0.031, respectively). Patients with DM had more maintenance renal replacement therapy and >10% decline in eGFR/year than patients without DM (*p* = 0.000, *p* = 0.000, respectively). No significant difference was observed in the body mass index (BMI) and serum albumin level between the two groups.

Patients with autoimmune disease had lower serum albumin levels and higher UPCRs compared to those without autoimmune disease (*p* = 0.005 and *p* = 0.019, respectively).

### 3.2. Baseline Renal Ultrasonography Findings and Differences between Patients with and without DM

Imaging findings of patients are shown in Table 2. The mean longitudinal right kidney size was 10.45 (range, 7.8–13.4) cm and the mean RL/height (RH) was 6.44 (5.28–8.99, cm/m). The mean parenchymal thickness (PK) and CK were 1.42 (range, 0.84–2.45) cm and 0.66 (range, 0.31–1.41) cm. The mean PK/kidney/height (PKH) and CK/kidney/height (CKH) were 0.88 (range: 0.56–1.45) and 0.41 (range: 0.21–0.87) cm, respectively (Table 3).

Patients with DM had higher PKH (*p* = 0.006) than those without DM, despite having lower eGFR. No significant difference was observed in RL, RH, and CKH between the two groups. No significance differences were observed in renal ultrasound findings between patients with and without autoimmune disease.

### 3.3. Association between Renal Ultrasonography Measurements and Baseline Clinical Parameters

The relationship between renal ultrasonography findings and baseline clinical parameters was assessed. The RH was significantly correlated with eGFR at the time of renal ultrasonography (*p* = 0.000, Figure 2). The mean CK and PK were also significantly correlated with eGFR at the time of renal ultrasonography (*p* = 0.000, *p* = 0.003, respectively; Figure 3B,C).

In both patients with and without DM, a significant positive correlation was observed with height and BMI for RL; however, a significant negative correlation was observed with age. Renal echogenicity was also significantly negatively correlated with eGFR and positively with urine protein.

### 3.4. Association between Clinical and Ultrasonography Parameters and Renal Prognosis

We evaluated the association between renal prognosis and clinical parameters and found that 33 patients (13.1%) experienced renal disease progression (≥10% decline in eGFR/year or the initiation of renal replacement therapy) during the follow-up period. Patients were divided into two groups (progression and non-progression groups) based on whether the renal disease had progressed. In the progression group, the renal scoring system and the presence of DM and UPCR were significantly higher, whereas eGFR and serum hemoglobin and albumin levels were lower than in the non-progression group. Moreover, no other parameters including autoimmune disease showed statistically significant differences (Table 4).

In ultrasound findings, the RL, RH, CKH, and PKH levels were not significantly correlated with renal progression. The number of patients with elevated renal echogenicity levels was higher in the progression group than that in the non-progression group. In particular, the number of patients with higher renal echogenicity in the DM group was higher than that in the non-DM group.

### 3.5. Association between High Renal Scoring and Kidney Disease Progression

The presence of DM, higher renal scores and UPCR, lower hemoglobin and eGFR were significantly associated with renal disease progression in univariate analyses. All confounding variables, including age, the presence of DM, HTN, serum hemoglobin, eGFR, and UPCR, were included in multivariate logistic regression analysis to determine the independent effects of the renal scoring system on renal progression. In multivariate logistic regression analysis, higher renal scores, the presence of DM, and younger age were independently associated with renal disease progression (Table 5).

## 4. Discussion

In our study, the height-corrected kidney length (right side kidney), CK and PK, surface irregularity, and echogenicity alone were found to not significantly predict renal disease progression. However, the renal scoring system calculated using renal ultrasound findings was found to significantly predict renal disease progression. Furthermore, despite the renal function of patients with DM being poorer than that of patients without DM, they had similar kidney sizes and thicker mean CK and PK than those without DM.

The currently recommended screening for CKD in patients with DM is based on the measurement of albumin excretion rate and eGFR. However, eGFR is not useful for the diagnosis of glomerular HTN and renal hypertrophy, which are earlier findings of diabetic kidney disease.

In our study, patients with DM were older and had higher UPCR, more renal disease progression, and lower eGFR than those without DM. Renal size and volume were significantly decreased with age in patients without DM; however, no significant findings were observed in patients with DM [12], and another study reported that the renal volume of patients with DM was higher than that of healthy participants [16]. Our study showed that despite the low eGFR, mean CK and PK were thicker in patients with DM than in those without DM, and no differences were observed between the two groups in RL, irregularity, and echogenicity.

RL, measured by ultrasound, is the most commonly used surrogate marker to evaluate renal function. Kidney size has been considered as a factor associated with poor renal outcomes in the general population and patients with kidney disease. In the healthy population, a short kidney length has been independently associated with low BMI, short height, and high serum creatinine levels [17]. Other studies showed that RL is not significantly associated with the prediction of renal impairment [18]. Our study showed that RL was associated with height and BMI in patients with and without DM, and it positively correlated with eGFR and UPCR in both groups.

A shorter RL was also associated with increased cardiovascular risk in children [19] and was a significant predictor of mortality in patients with DM on peritoneal dialysis [20]. Compared to absolute RL, height-adjusted RL was considered as a better variable when sex and height variability were considered [21,22]. Therefore, we attempted to find the relationship between the absolute and relative RLs and renal disease prognosis; however, we could not find a significant association with the renal disease prognosis in patients with and without DM.

Our study also revealed that RL is significantly associated with initial renal function. Previous studies have reported that RLs are significantly correlated with eGFR [7,18,23]; however, other studies have shown that RL is not significantly associated with eGFR [24]. Based on these results, the kidney length (absolute or relative) may reflect the current renal function; however, kidney length alone is not related to renal disease progression.

A previous study showed that a thin, high echogenic cortex in ultrasonography refers to irreversible damage, whereas a thick, normal echogenic cortex may indicate reversible damage [11]. Libório et al. reported that a higher renal echogenicity (kidney/liver ratio > 1.15) can identify those with irreversible advanced CKD in patients with glomerular disease and a normal kidney size (≥8 cm) [25]. In our study, renal echogenicity was negatively correlated with eGFR and positively with UPCR. In addition, we showed that patients with high echogenicity were significantly associated with renal disease progression. Measurement of renal echogenicity could help evaluate the initial renal status and prognosis.

Ultrasonographic measurement of renal dimension, and, in particular the cortical thickness, suffers from high inter- and intra-observer variability, so it is not commonly used due to poor cortico-medullary differentiation [26]. In our study, in order to reduce the inter-observed variability for the measurement results between the three investigators, we calculated the average of the cortical and parenchymal thicknesses of the upper, middle, and lower poles of each height. In addition, CK/PK was used in the calculation formula of renal scoring to reduce the variability of measured values.

Renal CK measurement was considered an independent parameter of renal function impairment in patients with CKD [27], and serial renal CK progression was positively associated with eGFR during follow-up [28]. Renal PK to indicate the chronicity of renal failure [29], where PK was measured by computed tomography, was a significant predictor of relative renal function [30]. Our study also showed that mean renal PK was significantly correlated with the initial renal function. We also showed that the PKH of patients with DM is higher than of those without DM, despite decreased renal function.

The study’s findings suggest that it is difficult to predict the renal prognosis in patients with DM only with the findings such as, RL, mean PK and CK, irregularity, and echogenicity of renal ultrasound. Therefore, the new renal scoring system consists of renal parenchymal echogenicity, the shape of the cortical margin, right kidney length/height (RH), CK/kidney length/height (CKH), and CK/PK (PK), constructed as a new index. In this study, the new renal scoring system successfully confirmed an independent association with the prognosis of patients with DM, and an association with early GFR and proteinuria.

The renal resistive index (RI = peak systolic velocity − telediastolic velocity/peak systolic velocity) is widely used in sampling the interlobar arteries to assess parenchymal perfusion. The RI is increased in hypertensive nephropathy and is correlated with the histological severity of nephrosclerosis and with CKD in hypertensive patients. Furthermore, RI > 0.80, in the case of renal artery stenosis, has a negative prognostic value for revascularization. Glomerular hyperfiltration represents the first reversible stage of diabetic nephropathy leading to chronic kidney disease. RI is a possible marker of hyperfiltration in diabetic patients. RI value > 0.80 acts as a major prognostic sign for progression [31]. However, we could not include data on renal perfusion in the study because we did not measure the resistive index (RI) during renal ultrasound in patients, excluding renal transplant patients.

Our study had several limitations. First, our study had a relatively small number of patients, which strongly limits the statistical power. Second, a retrospective study design cannot exclude all confounding factors. Third, the accuracy of the new scoring system could not be confirmed by other statistical methods. Fourth, cortical echogenicity was evaluated visually rather than quantitatively.

In conclusion, the renal parenchymal thickness of patients with DM was significantly higher than that of those without DM, despite showing decreased renal function. Patients with high renal scores were significantly associated with renal disease progression. Our results suggest that renal ultrasonography at the time of diagnosis provides useful prognostic information in patients with kidney disease. Further large-scale studies are needed to establish the relationship between the renal scoring system and renal disease progression in patients with kidney disease.

## Figures and Tables

**Figure 1 diagnostics-13-00515-f001:**
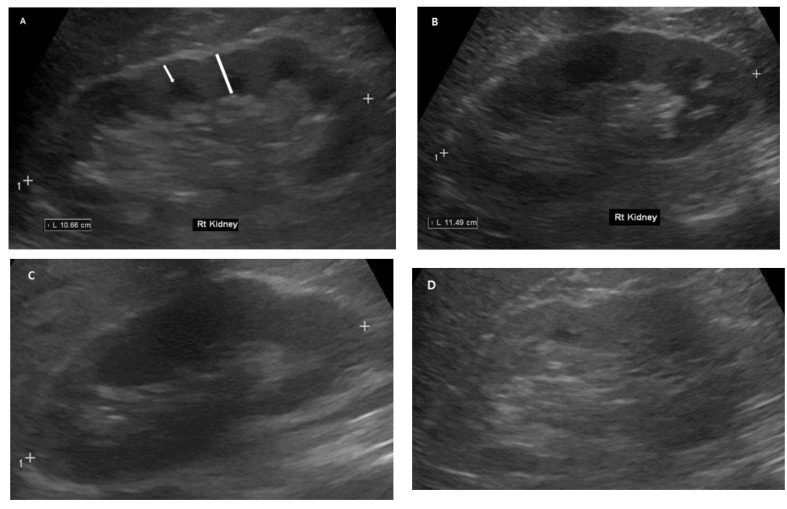
Representative pictures of ultrasound findings: (**A**) a 63-year-old male with normal renal ultrasound; (**B**) a 54-year-old female with diabetes; (**C**) a 54-year-old male with eGFR > 60 mL/min/1.73 m^2^; (**D**) a 65-year-old male with eGFR 45 mL/min/1.73 m^2^ (arrow: measurement of cortical thickness; bar: measurement of parenchymal thickness).

**Figure 2 diagnostics-13-00515-f002:**
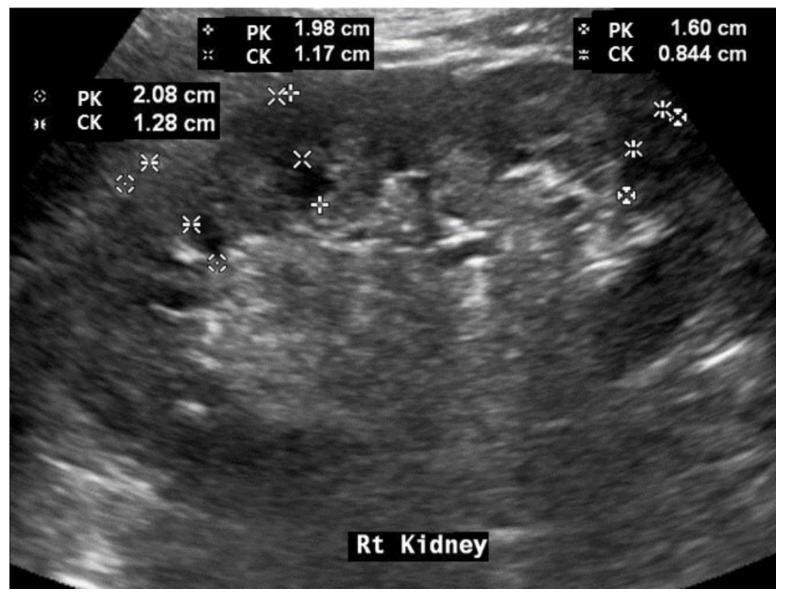
Representative picture of measurements using renal scoring.

**Figure 3 diagnostics-13-00515-f003:**
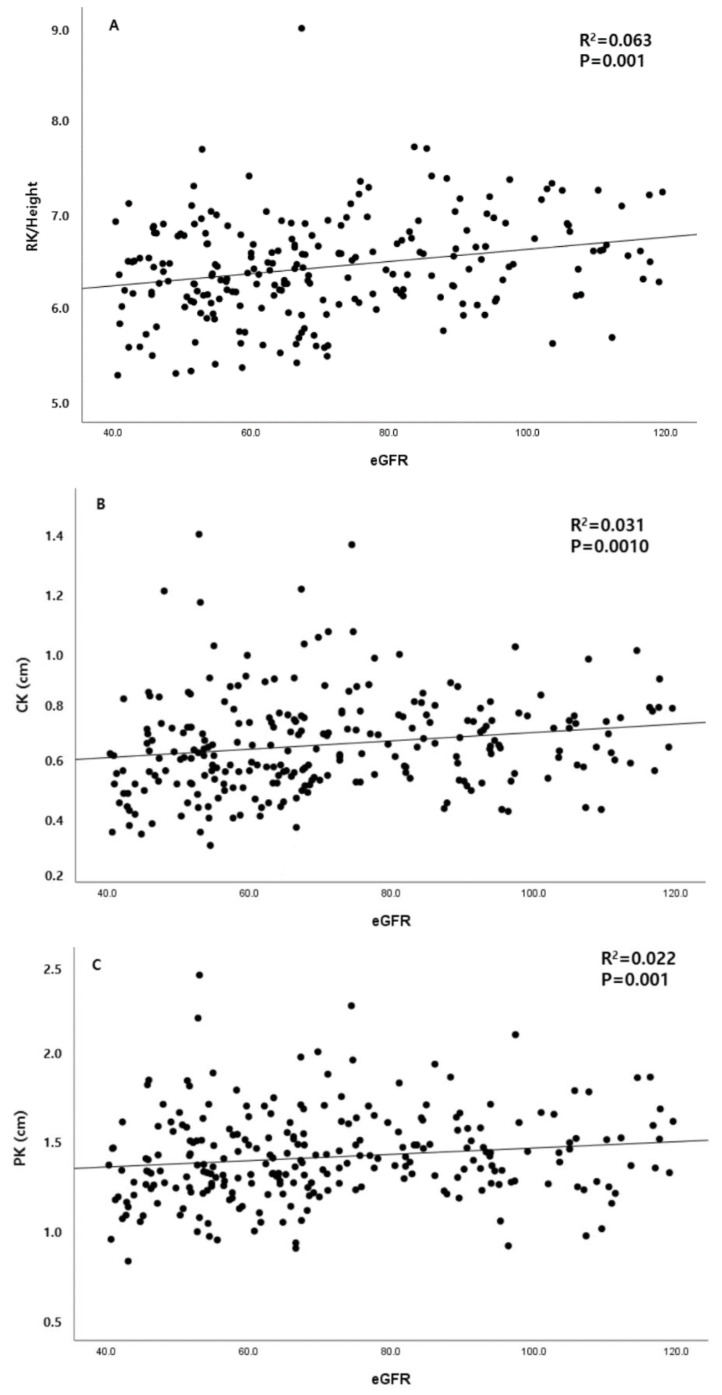
Correlation between ultrasound findings and baseline eGFR. (**A**) RH and eGFR. (**B**) CK and eGFR. (**C**) PK and eGFR. Abbreviations: eGFR: estimated glomerular filtration rate; RH: renal length/height; CK: cortical thickness; PK: parenchymal thickness.

**Table 1 diagnostics-13-00515-t001:** Renal scoring calculation for representative patient.

Value	Scoring	Value		Scoring
RK (cm)	11.6	Height (m)		1.698
CK (cm)	1.1	PK (cm)		1.89
RH: RK/Height	6.83 (Median: 6.427)	CKH, 10 × CK (RH)		1.61 (Median: 0.99)
value				Scoring
Echogenicity	Normal (0)	Slightly increased (1)	Increased (2)	0
Capsular irregularity	Normal (0)	Slightly irregular (1)	Irregular (2)	0
CK/PK	Above (0)	Median	Below (1)	0
RH	Above (0)	Median	Below (1)	0
CKH	Above (0)	Median	Below (1)	0
Total scoring				0

Abbreviations: RK: right kidney; RH: renal length/height; CK: cortical thickness; PK: parenchymal thickness; CKH: (cortical thickness/renal length)/height.

**Table 2 diagnostics-13-00515-t002:** Baseline characteristics of the study subjects.

	Total (N = 252)	Non-DM (149)	DM (103)	*p*-Value
Age (years)	59.72 ± 9.66 (27–81)	58.15 ± 10.24	61.99 ± 8.27	0.001
Male sex, n (%)	142 (56.1%)	80 (53.7%)	62 (60.2%)	0.186
HTN, n (%)	166 (65.6%)	80 (53.7%)	86 (83.5%)	0.000
BMI	25.1 ± 3.73 (16.2–39.1)	24.72 ± 3.80	25.56 ± 3.60	0.100
CKD stage	55:104:93 (21.8:41.3:36.9, %)	43:62:44 (28.9:41.6:29.5, %)	12:42:49 (11.7:40.8:47.6, %)	0.000
Laboratory parameters				
HbA1c	6.87 ± 1.76 (4.4–13.2)	5.49 ± 0.62	7.34 ± 1.78	0.000
Hemoglobin (g/dL)	13.2 ± 1.8 (9.3–18.7)	13.7 ± 1.68	12.5 ± 1.61	0.000
Albumin (mg/dL)	3.97 ± 0.59 (1.6–4.9)	4.01 ± 0.64	3.91 ± 0.53	0.194
Serum creatinine (mg/dL)	1.12 ± 0.34 (0.52–2.13)	1.05 ± 0.32	1.21 ± 0.34	0.000
eGFR (mL/min per 1.73 m^2^)	71.1 ± 20.7 (40.4–119.5)	75.2 ± 20.8	65.3 ± 19.1	0.000
Glucose (mg/dL)	132.1 ± 66.3 (42–536)	106.6 ± 22.3	169.3 ± 87.8	0.000
UPCR (g/g)	1.17 ± 2.20 (0.01–14.20)	0.75 ± 1.21	1.62 ± 2.82	0.031
LDL (mg/dL)	110.1 ± 48.3 (20–119)	123.6 ± 55.2	91.7 ± 28.5	0.000
ΔeGFR	−8.7 ± 25.3 (−109.6–101.1)	−1.5 ± 22.6	−19.2 ± 25.3	0.000
Slope of ΔeGFR (/y)	−2.12 ± 6.18 (−24.7–25.27)	−0.31 ± 5.56	−4.73 ± 6.11	0.000

Abbreviations: DM: diabetes mellitus; HTN: hypertension; BMI: body mass index; CKD: chronic kidney disease; eGFR: estimated glomerular filtration rate; UPCR: urine protein-to-creatinine ratio; LDL: low-density lipoprotein; ΔeGFR: eGFR change.

**Table 3 diagnostics-13-00515-t003:** Baseline ultrasound parameters of the study subjects.

	Total (N = 239)	Non-DM (149)	DM (103)	*p*-Value
RL (cm)	10.45 ± 0.94 (7.8–13.4)	10.37 ± 0.95	10.55 ± 0.93	0.141
CK (cm)	0.66 ± 0.17 (0.31–1.41)	0.65 ± 0.16	0.69 ± 0.19	0.049
PK (cm)	1.42 ± 0.24 (0.84–2.45)	1.38 ± 0.23	1.49 ± 0.27	0.001
CK/PK	0.46 ± 0.65 (0.29–0.70)	0.47 ± 0.06	0.46 ± 0.07	0.447
CK/RK	0.06 ± 0.01 (0.04–0.12)	0.06 ± 0.01	0.06 ± 0.02	0.074
PK/RK	0.14 ± 0.02 (0.08–0.21)	0.13 ± 0.02	0.14 ± 0.02	0.005
RH	6.44 ± 0.53 (5.28–8.99)	6.41 ± 0.54	6.48 ± 0.51	0.150
CKH	1.03 ± 0.26 (0.52–2.11)	1.38 ± 0.23	1.06 ± 0.28	0.147
PKH	2.22 ± 0.37 (1.45–3.47)	2.16 ± 0.36	2.29 ± 0.37	0.007
Irregularity (%)	0 (71.0%), 1 (22.6%) 3 (6.3%)	0 (75.8%), 1 (17.4%) 3 (6.7%)	0 (64.1%), 1 (30.1%) 3 (6.5%)	
Echogenicity (%)	0 (59.1%), 1 (29.8%) 3 (11.1%)	0 (57.7%), 1 (32.2%) 3 (10.1%)	0 (61.2%), 1 (26.2%) 3 (12.6%)	

Abbreviations: RL: absolute right renal length; CK: cortical thickness; PK: parenchymal thickness; RH: renal length/height; CKH, (cortical thickness/renal length)/height; PKH: (parenchymal thickness/renal length)/height.

**Table 4 diagnostics-13-00515-t004:** Comparison of clinical and ultrasound findings between progression and non-progression groups.

	Non-Progression (N = 219)	Progression (N = 33)	*p*-Value
Age (years)	59.94 ± 9.53	58.30 ± 10.49	0.366
Male sex, n (%)	13 (39.4%)	97 (44.3%)	0.707
DM, n (%)	76 (34.7%)	27 (81.8%)	0.000
HTN, n (%)	140 (63.9%)	26 (78.8%)	0.093
BMI	25.19 ± 3.82	24.39 ± 3.05	0.278
Laboratory parameters			
HbA1c	6.52 ± 1.50	8.26 ± 2.04	0.000
Hemoglobin (g/dL)	13.31 ± 1.79	12.38 ± 1.83	0.004
Albumin (mg/dL)	4.04 ± 0.58	3.54 ± 0.56	0.000
Serum creatinine (mg/dL)	1.09 ± 0.33	1.32 ±0.36	0.000
eGFR (mL/min per 1.73 m^2^)	72.62 ± 20.53	60.21 ± 17.67	0.001
UPCR (g/g)	0.86 ± 1.82	3.21 ± 3.20	0.001
LDL (mg/dL)	112.23 ± 50.03	96.83 ± 34.05	0.157
Ultrasound parameter			
RL (cm)	10.43 ± 0.94	10.54 ± 0.97	0.538
RH	6.45 ± 0.54	6.36 ± 0.47	0.373
CKH	0.41 ± 0.10	0.38 ± 0.12	0.170
PKH	0.88 ± 0.15	0.87 ± 0.13	0.576
Renal scoring system	2.27 ± 1.644	3.03 ± 1.794	0.016
Echogenicity	0:62.1%, 1:29.2%, 2:8.7%	0:39.4%, 1:33.3%, 2:27.3%	

Abbreviations: M: diabetes mellitus; HTN: hypertension; BMI: body mass index; CKD: chronic kidney disease; eGFR: estimated glomerular filtration rate; UPCR: urine protein-to-creatinine ratio; LDL: low-density lipoprotein; RL: absolute right renal length; RH: renal length/height; CKH: (cortical thickness/renal length)/height; PKH: (parenchymal thickness/renal length)/height.

**Table 5 diagnostics-13-00515-t005:** Logistic regression analysis for the occurrence of renal progression.

Factors	Univariate		Multivariate	
	HR (95% CI)	*p*-Value	HR (95% CI)	*p*-Value
Scoring system	1.599 (1.021–1.530)	0.016	1.413 (1.032–1.933)	0.031
Age (Years)	0.983 (0.946–1.020)	0.365	0.919 (0.864–0.978)	0.007
DM (n, %)	8.467 (3.350–21.40)	0.000	4.917 (1.325–18.25)	0.017
HTN (n, %)	2.096 (0.870–5.048)	0.099	1.215 (0.290–5.081)	0.790
Hemoglobin (g/dL)	0.717 (0.568–0.905)	0.005	0.848 (0.605–1.188)	0.337
Albumin (mg/dL)	0.339 (0.201–0.573)	0.000	0.396 (0.135–1.163)	0.396
eGFR (mL/min/1.73 m^2^)	0.964 (0.941–0.986)	0.002	0.971 (0.938–1.002)	0.094
UPCR (g/g)	1.388 (1.187–1.623)	0.000	1.238 (0.962–1.593)	0.097
Autoimmune disease (n, %)	0.364 (0.083–1.593)	0.179	0.204 (0.025–1.674)	0.204
RH (cm)	1.129 (0.768–1.659)	0.536		
CK (cm)	0.855 (0.101–7.207)	0.886		
PK (cm)	2.267 (0.556–9.241)	0.253		
CKH	0.608 (0.047–5.964)	0.608		
PKH	1.912 (0.377–9.711)	0.434		

## Data Availability

The data presented in this study are available on request from the corresponding author. The data are not publicly available due to privacy and ethical.

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
