# Peer review of "Ultrasound Renal Score to Predict the Renal Disease Prognosis in Patients with Diabetic Kidney Disease: An Investigative Study"

_diagnostics, 2023, doi:10.3390/diagnostics13030515_

Round 1

Reviewer 1 Report

Authors developed a renal scoring system calculated using renal ultrasound findings, that was found to significantly predict renal disease progression. The new renal scoring system was used in a limited population with DM and in a controll group. The score system consists of renal parenchymal echogenicity, the shape of cortical margin, right kidney length/Height (RH), CK/Kidneylength/height (CKH), PK/kidney/height (PKH), using standard abdominal probe ultrasound.

I have some concern on methodology and suggestions to improve the paper:

1    Title is be misleading for the reader. I suggest as example: ultrasound renal score to predict the renal disease prognosis in patients with Diabetic Chronic kidney disease: Investigative study

2   Methods to replicate this study are not well explained (Line 112-118)  Authors should be clearly write how is possible to calculate the score. I suggest to include a table or a picture how is possible to calculate the score based on one case with real measurement

3    A further limitation of the study is renal perfusion. The renal resistive index (RI=peak systolic velocity – telediastolic velocity/peak systolic velocity) is widely used sampling the interlobar arteries to assess parenchymal perfusion. The RI is increased in hypertensive nephropathy and correlates to the histological severity of nephrosclerosis and to CKD in hypertensive patients. Furthermore, RI >0.80 in the case of renal artery stenosis has a negative prognostic value for revascularization. Glomerular hyperfiltration represents the first reversible stage of diabetic nephropathy leading to chronic kidney disease. RI is possible marker of hyperfiltration in diabetic patients. RI value >0.80 acts as a major prognostic sign for progression. (cite Granata A, Galeano D, FioriniF. Acute and Chronic Nephropathy. Cap 2. In Martino P, Galosi AB (eds.): Atlas of Ultrasonography in Urology, Andrology, and Nephrology, DOI 10.1007/978-3-319-40782-1_2. Springer)

4   According to KDIGO’s guideline definition, chronic kidney disease progression is the clinical syndrome characterized by abnormalities of kidney structure or function, present for>3 months.

5   Did you measured inter-observer variability (among 5 operators)? US Measurement of renal dimension and in particular the cortical thickness suffers from high inter and intraobserver variability, so it is not commonly used due to poor cortico-medullary differentiation (cite Martino P, et al ; Imaging Working Group-Societa Italiana Urologia (SIU); Società Italiana Ecografia Urologica Andrologica Nefrologica (SIEUN). Practical recommendations for performing ultrasound scanning in the urological and andrological fields. Arch Ital Urol Androl. 2014 28;86(1):56-78. doi: 10.4081/aiua.2014.1.56. PMID: 24704936). In the text should be clearly that the median value of 3 measurement was used to reduce the intra-observed variability.

6  Abstract is too long and not interesting to read. 

Author Response

Thank you very much for your review. We will send you a correction for what you said. Please see the attachment.

Reviewer 2 Report

The paper is interesting and well written. I suggest to briefly discuss the impact of renal ultrasonography in patients with autoimmunune diseases and particularly if it correlates with the expression of inflammatory or immuinomodulating cytokines as TNF alpha and IL-17, proangiogenetic molecules as and VEGF (see and add as references papers by Murdaca et al concerning these cytokines and molecules)

Author Response

(The authors gave the same response as above.)

Round 2

Reviewer 1 Report

Authots addressed most of my concerns and I congratulate with them.

However referecnces and literature recall is not updated as requested in my previous comments 

Author Response

Thank you very much for your review. We will send you a correction for what you said. Please see the attachment.

Response to Reviewer 1 Comments

Authors addressed most of my concerns and I congratulate with them.

However referecnces and literature recall is not updated as requested in my previous comments 

Point 1: Authors addressed most of my concerns and I congratulate with them.

However, references and literature recall is not updated as requested in my previous comments.

Response 1:

Thank you for the valuable comment. Based on your opinion, we have inserted contents and references related to renal perfusion into the discussion part.

⇒ The renal resistive index (RI=peak systolic velocity – telediastolic velocity/peak systolic velocity) is widely used sampling the interlobar arteries to assess parenchymal perfusion. The RI is increased in hypertensive nephropathy and correlates to the histological severity of nephrosclerosis and to CKD in hypertensive patients. Furthermore, RI >0.80 in the case of renal artery stenosis has a negative prognostic value for revascularization. Glomerular hyperfiltration represents the first reversible stage of diabetic nephropathy leading to chronic kidney disease. RI is possible marker of hyperfiltration in diabetic patients. RI value >0.80 acts as a major prognostic sign for progression (30). However, we could not include data on renal perfusion in the study because we did not measure resistive index (RI) during renal ultrasound in patients other than renal transplant patients.

 (Page7 Line 265-274)

(Reference)

  1. Granata A, Galeano D, Fiorini F. Acute and Chronic Nephropathy. In: Martino P, Galosi AB, editors. Atlas of Ultrasonography in Urology, Andrology, and Nephrology. Cham: Springer International Publishing; 2017. p. 13-26.

⇒ Ultrasonographic measurement of renal dimension and in particular the cortical thickness suffers from high inter and intra-observer variability, so it is not commonly used due to poor cortico-medullary differentiation (25). In our study, in order to reduce the inter-observed variability for the measurement result between three investigators, we calculated the average of the cortical and parenchymal thickness of the upper, middle, and lower pole of each height. In addition, CK/PK was used in the calculation formula of renal scoring to reduce the variability of measured values.

(Page 6. Line 244-250)

(Reference)

  1. Martino P, Galosi AB, Bitelli M, Consonni P, Fiorini F, Granata A, et al. Practical recommendations for performing ultrasound scanning in the urological and andrological fields. Arch Ital Urol Androl. 2014;86(1):56-78.